# A Novel Chitosan Composite Biomaterial with Drug Eluting Capacity for Maxillary Bone Regeneration

**DOI:** 10.3390/ma16020685

**Published:** 2023-01-10

**Authors:** Barbara Giordano-Kelhoffer, Raquel Rodríguez-Gonzalez, Marina Perpiñan-Blasco, Jenifer O. Buitrago, Begoña M. Bosch, Roman A. Perez

**Affiliations:** 1Bioengineering Institute of Technology, Universitat Internacional de Catalunya (UIC), 08017 Barcelona, Spain; 2Faculty of Dentistry, Universitat Internacional de Catalunya (UIC), 08017 Barcelona, Spain; 3Faculty of Medicine and Health Sciences, Basic Science Department Universitat Internacional de Catalunya (UIC), 08017 Barcelona, Spain

**Keywords:** bone graft, chitosan, hydroxyapatite, doxycycline

## Abstract

Bone grafting is one of the most commonly performed treatments for bone healing or repair. Autografts, grafts from the same patient, are the most frequently used bone grafts because they can provide osteogenic cells and growth factors at the site of the implant with reduced risk of rejection or transfer of diseases. Nevertheless, this type of graft presents some drawbacks, such as pain, risk of infection, and limited availability. For this reason, synthetic bone grafts are among the main proposals in regenerative medicine. This branch of medicine is based on the development of new biomaterials with the goal of increasing bone healing capacity and, more specifically in dentistry, they aim at simultaneously preventing or eliminating bacterial infections. The use of fibers made of chitosan (CS) and hydroxyapatite (HA) loaded with an antibiotic (doxycycline, DX) and fabricated with the help of an injection pump is presented as a new strategy for improving maxillary bone regeneration. In vitro characterization of the DX controlled released from the fibers was quantified after mixing different amounts of HA (10–75%). The 1% CS concentration was stable, easy to manipulate and exhibited adequate cuttability and pH parameters. The hydroxyapatite concentration dictated the combined fast and controlled release profile of CSHA50DX. Our findings demonstrate that the CS-HA-DX complex may be a promising candidate graft material for enhancing bone tissue regeneration in dental clinical practice.

## 1. Introduction

One of the most common bone healing procedures is bone grafting. [1]. Infection, trauma, tumor, congenital etiology, and surgery are all potential causes of bone defects [2]. To address this issue, bone grafting can be accomplished by using material extracted from the patient, as well as synthetic or natural material substitutes. Bone grafting has grown dramatically in the last two decades as a result of advances in material design as well as due to the development of new grafting procedures, and it is expected to grow further in the future [3]. The main reason is an increase in tooth loss in the population as a result of caries, periodontal disease, dental trauma, or tumors [4], which cause alveolar bone loss which eventually jeopardizes dental implant success. Consequently, bone grafting in the oral cavity is mainly used prior to the placement of dental implants.

Autografts are the gold standard for bone regeneration, but they have limited availability. Natural polymers have recently gained popularity as a result of the search for alternative sources for bone regeneration [5,6,7]. Some of the most researched natural polymers for bone regeneration are collagen, hyaluronic acid, alginate, and chitosan. Furthermore, natural polymers are used in medicine, specifically for the maintenance of oral health. One of their main assets is their biocompatibility, which allows them to elicit a specific response from the body while not interfering with the immune system [8]. Various applications have also been discovered in the pharmaceutical and biomedical fields, particularly as drug-delivery systems, wound dressing, and scaffolds for tissue engineering, among many others [7,9,10]. Some of these biomaterials are pH sensitive due to their chemical composition and have been found to be advantageous because they can deform or degrade in acidic or alkaline environmental conditions, enabling the creation of tailor-made materials that are ideal as drug carriers [11]. Among these, chitosan (CS)-based biomaterials are an appealing option because they are derived from chitin, which is currently the second-most abundant polysaccharide used, is highly available in nature, and is renewable [5,12,13,14]. Moreover, CS is a natural cationic polymer that exhibits good biodegradability, biocompatibility, and non-toxic properties [10]. Because CS is positively charged, it can bind to the negatively charged bacterial cell wall. Hence, it can attach to the DNA, preventing bacterial replication [15,16]. The antibacterial property of CS distinguishes chitosan as a distinct material, inhibiting the growth of a wide variety of fungi, yeasts, and bacteria, and thus making CS beneficial for use in the dentistry field [17,18,19,20].

Interestingly, CS can be molded into different forms to accommodate various maxillary bone defects. As a result, CS fibers are excellent candidates for resolving various bone regeneration scenarios. Nevertheless, a pure CS graft is mechanically weak and lacks the bioactivity to promote hard tissue formation, limiting its use as a bone graft biomaterial [21,22]. Polymer–ceramic composite bone grafts have recently been extensively researched, showing promise in mimicking the organic and inorganic components of natural bone. The use of a chitosan–hydroxyapatite composite improved in vitro mechanical properties and bioactivity [23,24]. Hence, hydroxyapatite (HA) appears to be an interesting ceramic counterpart based on its biocompatibility, low cost, and osteogenic properties [24]. Moreover, this ceramic has a chemical composition that is similar to the mineral phase of natural bone, which is calcium phosphate, and thus has bone-bonding ability [25,26,27,28]. There are several options for treating maxillary bone loss. Enhancing maxillary bone regeneration with biomaterials is one of them, and it has gained popularity in recent years, particularly through the use of bioceramics, which are thought to be a good option due to the properties mentioned above. However, because every maxillary bone defect is different in shape and size, using bioceramic materials alone may be difficult for the clinician to adapt correctly. Because of its easily tailorable properties, a natural polymer has been chosen as the matrix of the composite. Furthermore, natural polymers have excellent biocompatibility. Among natural polymers, chitosan has grown in popularity in tissue engineering (FDA approval for wound dressings [29]) and drug delivery. We chose it to synthesize our fibers because it is a biodegradable, non-toxic, and mucoadhesive biomaterial [30], with a low-cost and sustainable processing.

In this study, a chitosan–hydroxyapatite-based bone graft biomaterial loaded with an antibiotic molecule was designed to produce an improved biomaterial that is easy to fabricate, manipulate, and individualize for use in the dentistry field. Nonetheless, bone grafting procedures may result in bacterial infections, which may compromise tissue regeneration outcomes. To avoid these infections, local drug release is an excellent strategy for preventing general adverse effects by shielding surrounding tissues from rapid drug exposure while also improving drug efficacy by achieving controlled release directly at the infection site. This procedure may also benefit patients by avoiding unnecessary systemic undesirable effects, because the drug will not pass through the gastrointestinal barrier. Additionally, it will be more efficient and easier for the patient, avoiding possible forgetfulness of the patient’s drug intake. Antibiotic loading on bone-regenerative biomaterials is a promising method of preventing infection during the resorption stage of bone substitutes throughout augmentation procedures [31]. Doxycycline (DX) was therefore combined with the CS–HA composite biomaterial. It is a tetracycline antibiotic with a broad spectrum of action that has been used to treat bacterial infections of the oral cavity [32]. Besides that, tetracyclines, together with other antibiotics such as metronidazole, have been reported to be the only antibiotics that stimulate bone mineralization [33,34]. To stimulate maxillary bone regeneration, we propose a simple method involving the use of an injection pump to fabricate chitosan fibers with the incorporation of HA and DX. Hence, the goal of this study is to characterize the various fibers obtained and to study the DX-controlled release while comparing different percentages of HA.

## 2. Materials and Methods

### 2.1. Pristine Chitosan Fibers Synthesis and Characterization

CS fibers were designed as follows: Fibers were prepared in order to either form fiber-based scaffolds for tissue engineering or to determine the optimum conditions for future use in 3D printing. First, CS (Sigma-Aldrich, St. Louis, MO, USA) was prepared by dissolving 1 mL of glacial acetic acid (CH3COOH) (Panreac, Barcelona, Spain) into 99 mL of deionized water (DI water, from the Milli-Q water system, Merck, Barcelona, Spain) and stirring at 800 rpm. The solution was then treated with varying amounts of chitosan, ranging from 1 to 3 g, and stirred at 450 rpm for one hour at room temperature, to produce 1 to 3% wt chitosan solutions. Next, a volume of 3 mL of CS solution was introduced into a 5 mL plastic syringe. Different parameters were varied to obtain fibers with different sizes, textures, and consistencies. The syringe was then placed in an injection pump (KDS-200-CE, KD Scientific Inc., Holliston, MA, USA) which was then used to extrude the fibers at different constant injection speeds, ranging from 60 to 150 mL/h using a needle diameter of 0.5 or 0.9 mm. The fibers were extruded into an NaOH solution, which ranged from 0.05 M to 0.5 M and was used as the crosslinking agent. Fibers were left for 5 min to complete the crosslinking. Hence, three main variables were studied: injection speed, needle diameter, and crosslinking solution. The samples were fabricated in triplicates. The collection of fibers from the NaOH solution was made with tweezers.

### 2.2. Fiber Structure Analysis

Once the CS fibers were prepared, their stability and manipulation were evaluated under qualitative parameters to compare between the different variables studied. The maintenance of the 3D structure of the fibers was evaluated and scored as low or bad, resulting in a non-manipulable fiber that may easily lose its structure when handling with tweezers. A correct designation meant the fiber was manipulable but not stable in aqueous media. An optimal designation meant the result was an excellently manipulable and stable fiber in aqueous media and maintenance of its structure when handling with tweezers. An optical microscope (Olympus CKX41, Nikon, Tokyo, Japan) was used to characterize the structure and to measure the diameter of the fabricated biomaterials.

### 2.3. pH Evolution

Considering the high pH of the crosslinking solution and in order to allow the fibers to exhale the excessive hydroxyl groups, the fibers were submitted to 4 washings for 10 min each. For this purpose, each individual fiber was immersed in 20 mL of deionized water to neutralize the basic pH of the solution (n = 3). The pH of the supernatant was measured using a pH-meter (SensION^TM^ + PH31) during the rinsing process. Fibers were considered rinsed once the pH of the supernatant reached a stable pH of 7.4.

### 2.4. Chitosan-Hydroxyapatite Composite Synthesis and Characterization

The composite fibers were prepared in a similar way to those described with the pristine chitosan fibers. In this case, different amounts of HA were incorporated into the chitosan mixture.

The 1% CS composite was used, and different amounts of HA were incorporated. First, hydroxyapatite (HA) powder (Sigma-Aldrich, St. Louis, MO, USA) was mixed with the CS solution at ratios of 10, 50, or 75 wt % of HA with respect to the composite to obtain CSHA biomaterials. Since the viscosity of CS varied with the HA concentration, the quantity of water used to mix with the HA powder was adjusted to the minimum value of water at which HA75% was well dispersed. HA 10, 50, and 75% were dispersed in 400 µL DI water and placed on the vortex to homogenize according to this criterion. A volume of 3 mL of CSHA composite was introduced on a 5 mL plastic syringe to fabricate the fibers with the injection pump, as explained before. A constant flow rate of 75 mL/h was established, and the needle diameter used was 0.9 mm. The crosslinking solution used was 0.5 M NaOH as the fibers crosslinking agent. All samples were fabricated in triplicates.

Once the CSHA fibers were manufactured, the characterization of their structure and the measurement of their diameter were evaluated using an optical microscope, as described for the pristine fibers.

The pH evolution experiment was assessed with the same steps followed for the CS fibers.

### 2.5. Drug Release from Composite Fibers

DX was selected to be incorporated into the CSHA composite because it is a broad-spectrum antibiotic and is known to have an osteogenic regeneration capacity [32,33]. The amount of DX concentration used to fabricate the fibers was 833 µg/mL. This concentration was selected taking into account a study that was similar to the present study, in which the researchers used 1 mg DX/mL of the material [24]. Thus, the antibiotic was added to the solution (either 1% CS or 1% CSHA composites) following the steps previously explained and using the same HA concentrations (0–75 wt %). To obtain CSHADX composites, a constant amount of doxycycline (DX) powder (Sigma-Aldrich, St. Louis, MO, USA) weight of 0.0025 g was added to the CSHA composite. In order to prevent light-induced oxidation, the DX-loaded biomaterials were kept in the dark until used. A volume of 3 mL of CSHADX composite was introduced on a 5 mL plastic syringe to fabricate the fibers with the injection pump, as explained before. A constant flow rate of 75 mL/h was established, and the needle diameter used was 0.9 mm. The crosslinking solution used was 0.5 M NaOH as the fibers’ crosslinking agent. All samples were fabricated in triplicates. The CSHADX composite structure analysis and pH evolution were assessed as explained before for the CS fibers.

### 2.6. In Vitro Doxycycline Release Study

After crosslinking the samples for 5 min in the 0.5 M NaOH solution, the amount of DX was analyzed from the supernatant to detect the amount of DX that had been released during the crosslinking and hence determine the amount of DX remaining in the fiber. The amount of DX released was measured using a spectrophotometer by measuring the absorbance at 351 nm (Bio-Tek Synergy HT, Winooski, VT, USA) [35]. Moreover, the effect of the HA amount in the mixture solution with CS was also investigated. To study the release of DX from the biomaterials after the crosslinking time, each sample was immersed in 1 mL DI water for different periods and maintained in the incubator at 37 °C. The released quantity was interpreted after normalizing to the loaded quantity. Then, 1 mL of medium was refreshed at each time point of the assay.

### 2.7. Statistical Analysis

Statistical analysis was carried out with a significance level of 5%. A Mann–Whitney test was conducted. Data are expressed as mean ± standard deviation. The results were statistically analyzed using Minitab (Minitab^®^ Software Inc. 17.1.0, State Collage, PA, USA) and GraphPad (GraphPad Prism Software Inc. version 6, San Diego, CA, USA) was used to graph the data.

## 3. Results and Discussion

### 3.1. Pristine Chitosan Fibers Fabrication

#### 3.1.1. Pristine Chitosan Fibers Characterization

We wanted to investigate the effect of various parameters on fiber fabrication, specifically CS concentration, flow rate, needle diameter, and NaOH molarity, in order to determine the best combination for potential composite fabrication. We analyzed the stability and manipulability of the fibers according to NaOH molarity. NaOH is a basic agent that allows for the physical crosslinking and gelation of CS. Chitosan is a glucosamine polysaccharide that has an amide linkage with acetic acid and contains a small proportion of amide groups [28]. Crosslinking is a physiochemical technique that involves a charged polymer, in this case CS, forming intermolecular chemical bonds between polymer chains to increase stability and resistance [7]. A sodium hydroxide solution (NaOH) was used to achieve CS crosslinking to form the fiber.

Three different NaOH molarities were investigated in order to select the optimal molarity: 0.05, 0.1, and 0.5 M. Low stability and manipulation of the CS fibers were obtained at lower molarities (0.05 and 0.1 M) during the working time studied. In contrast, the optimal molarity to ensure good stability and manipulation ability while maintaining the 3D fiber structure was 0.5 M (Appendix A). Because of the higher concentration of OH, this was the highest molarity tested and may yield the best results. These ions interact with CS cationic groups via a diffusion gradient, penetrating the composite quickly, allowing for rapid CS neutralization and the formation of a stable fiber network. As a result, the fiber production time can be effectively reduced [29]. Our findings are consistent with those of Bergonzi et al., who investigated different crosslinking media at different molarities and obtained good fiber constructs ranging from 0.5–1 M. In fact, they did not achieve 3D structural stability at lower crosslinking molarities, such as 0.1 M [29]. Furthermore, when compared with other studies that used NaCl or phosphate buffered saline, using a basic solution like NaOH for the crosslinking process of CS under mild conditions requires a shorter gelation time [36]. The ideal was to obtain stable fibers with the least amount of OH incorporated into the CS. The goal is to obtain fibers that are easy to manipulate at physiological pH levels. Some authors have described lengthy gelation processes that produced gels that dissolved in 24 or 48 h. However, given the possibility of a clinical application, this timing may not be ideal [29,36].

After determining that NaOH 0.5 M was the best cross-linking concentration, the effect of the needle diameter, polymer concentration, and flow rate was investigated. The needle diameter was determined by testing the stability and manipulability of CS fibers. Interestingly, the needle with the largest diameter (0.9 mm) provided the best stability and manipulation ability (Appendix A). We also investigated the effect of various CS concentrations (1, 2, and 3%) extruded at different flow rates (60, 75, 100, and 150 mL/h). When manipulated with tweezers, all CS concentrations demonstrated good fiber forming ability, which means they did not disintegrate and maintained their 3D structure. Flow rates at 60 mL/h, on the other hand, were less stable than higher rates (Appendix A). A wet spinning process was used to create the fibers, which included preparing the polymer and extruding it into the crosslinker agent for fiber formation. The polymer concentration is determined by its solubility and spinning pressure limitations. Fiber formation is affected by bath composition, temperature, and flow rate, among other factors [37], and flow rate is affected by needle diameter and polymer viscosity [38], which is related to concentration. The disintegration of very thin fibers obtained at 1% CS and 60 mL/h may be caused by a lack of crosslinking capacity at low rates [39]. This could be explained by the fact that using a low flow rate can result in poor polymer extrusion. This result is consistent with the findings of Pati et al., who found that CS fibers can be formed at higher rates [40]. Furthermore, the flow rate, in addition to the polymer concentration, can define the diffusion of NaOH to the fibers, which is higher as the polymer concentration increases, giving more fiber stability because it can reorganize chains [41]. A larger needle diameter may be preferable because it allows for better polymer flow through the needle. This ability of the polymer to flow may be related to how fast and far individual CS chains can move in relation to each other, as well as in fiber diameter, defining its stability and manipulation ability [38]. Figure 1 shows optical microscope images of fibers with different needle diameters, polymer concentrations, and flow rates. The pattern of 1% CS fibers extruded by the smallest needle diameter was smoother than that of higher CS concentrations extruded at the largest needle diameter. This is due to the low polymer content of the solutions. At same needle diameters, 1% CS fibers were thicker than higher CS%. Using different flow rates had no effect on the results. In addition, due to the high CS viscosity. We were unable to obtain fibers with the highest CS concentration (3%) and the lowest rate (60 mL/h); instead, we obtained beads.

We then wanted to quantify the different fibers, so we used image analysis software to analyze the images. We quantified fiber size by comparing two different variables simultaneously. We first investigated the impact of needle size and flow rate. Figure 2a showed that when the largest needle diameter (0.9 mm) was used, fibers significantly increased in size, nearly doubling the value when compared to smaller needle diameters (0.5 mm). In terms of the use of the different flow rates, we observed that there were no differences in fiber size. This finding could imply that only the needle diameter influences fiber size. A study demonstrated that when a larger volume of solution is in motion over a set distance, fibers are thicker [38]. We also investigated different CS concentrations at various flow rates using a needle diameter with a constant diameter of 0.9 mm (Figure 2b). When compared to other concentrations, the 1% CS concentration produced the highest fiber size values, which were statistically significant. At 1% CS concentration, there were no differences between the different flow rates. According to our findings, 1% CS had the largest fiber size. As a result, we hypothesize that fabricating the fibers at a lower CS concentration allowed for more polymer expansion within the crosslinking solution, resulting in thicker fibers. In this regard, a previous study found a non-linear relation between polymer concentration and fiber diameter in nanofibers [42]. Because the ultimate goal of developing this material is to fill bone defects, its stability and manipulation ability are critical to investigate; these characteristics, as well as the fiber shape, may allow for optimal manipulation, reduce treatment time, and improve material adaptation to the host tissue. Following our analysis, 1% CS fibers were chosen as the best for this purpose due to their having the widest size. This translates to using less material to fill a bone defect and having good cutting ability to make individualized shapes and sizes.

#### 3.1.2. pH Evolution

CS is a cationic polymer that is pH sensitive [43]. An excess of hydroxyl groups in the chitosan could be harmful once implanted because these hydroxyls may locally increase the pH and thus cause cell death. For this purpose, we wanted to ensure that we could neutralize the excess of hydroxyl groups through several rinsing processes. To determine the pH of our fibers, we immersed them in DI baths four times and analyzed them at different time points. We first wanted to see what difference the needle diameter (0.5 and 0.9 mm) made in 1% CS at 75 mL/h. A lower mean pH (8.4) was found at the lowest needle diameter compared to the highest needle diameter (8.7), with no significant differences after 48 h of the assay (Figure 2c). We also analyzed the effect of different CS concentrations (1–3%) at a constant flow rate (75 mL/h). After 24 and 48 h, the 1% CS sample had slightly higher pH values (8.7) than the 2 and 3% CS samples (8.6 and 8.5, respectively), with no significant differences (Figure 2d). After 48 h, all samples reached a pH near 7.4. Flow rates were also analyzed at 1% CS, with the lowest rate (60 mL/h) having the highest pH (8.9) and the highest rate (150 mL/h) having the lowest pH (8.2) with no statistical significance at all time points analyzed (Figure 2e). As we have seen, pH influences CS behavior, which is due to the amino groups in its chains. When it reaches a weak acidic pH (<6), the protonated free amino groups of glucosamine allow for its solubility; on the other hand, a basic solution such as NaOH (pH = 12) allows for the formation of CS fibers. This is a common approach for performing the sol-to-gel transition for water-soluble, pH-sensitive polymers. Since the functional groups on the polymer either accept or donate protons as a result of pH changes in the environment, CS typically undergoes phase transition [44]. Because CS fibers were produced in NaOH and had a pH = 12, we immersed them in several DI water baths to lower the pH and change it to a physiological pH. For this reason, we wanted to assess the pH stability of CS fibers before proceeding with further experiments. Considering that there were no significant differences between the studied variables, we chose 1% CS, 75 mL/h, and 0.9 mm for further analysis.

### 3.2. Composite Fibers

#### 3.2.1. Composite Fibers Characterization

The morphology of the CS and CSHA fibers obtained by means of coagulating in the NaOH solution is shown in Figure 3a. The composite macrostructure becomes more uniform and condensed as the HA concentration increases. The explanation could be that the greater the quantity of HA, the less water there is. As a result, the material is less hydrogel and more ceramic. Moreover, higher HA concentrations turned the fibers a whitish color. Optical microscope images of the composite fibers revealed the individual structures of the different samples analyzed. The CS and CSHA75 fiber sizes appear to be the thickest. Figure 3b shows the different fiber sizes. When compared to the groups with HA incorporation, the raw CS sample had significantly higher fiber size values. CSHA75 had the largest fiber size among CSHA samples, which was statistically significant.

#### 3.2.2. pH Evolution

The effect of pH showed no statistically significant differences between the groups studied. Nonetheless, increasing the HA concentration in CS composites resulted in a direct proportional correlation with decreasing pH values (Figure 3c and Figure 4c). After 72 h, the pH of the analyzed composites was close to 7.4.

There are several factors that may affect the pH value. One of the most significant is the absorption relationship between anions and cations. In general, higher cation absorption causes lower pH, while higher anion absorption causes higher pH. When CS is combined with HA, the ion pairs formed by negatively charged phosphate groups of HA and protonated amine functionality of CS in the crosslinking process [45] may allow for greater absorption of cations in the media, resulting in a decrease in pH.

#### 3.2.3. Drug Loading Release Capacity from Composite Fibers

DX is an inexpensive broad-spectrum antibiotic used to treat bacterial infections caused by aerobic and anaerobic Gram-positive and Gram-negative bacteria, as well as other microorganisms. Its antimicrobial activity is attributed to the inhibition of bacterial protein biosynthesis. DX also inhibits collagenase activity, preventing osteoclasts from resorbing bone [46,47,48].

Figure 4 shows DX-incorporated CS and CSHA fibers. In these samples, a similar trend to that seen in CSHA fibers was observed, with a uniform and better condensed macrostructure at higher amounts of HA, but less whitish color due to the yellowish color of the DX powder. The optical microscope images revealed a well-defined fiber shape, with CSHA75DX being the thickest composite (Figure 4a). According to the optical microscope images, CSHA75 and CSHA50 had similar sizes and were larger than the other two conditions. When the fibers are compared to each other, it is clear that the addition of DX has no effect on the fiber size in the HA10 condition (Figure 4b). It should be noted that CS composites were previously mixed with 400 µL of DI water to disperse DX particles. This may explain the differences in fiber size between CS and CSHA composites with DX incorporation.

Figure 4c compares the pH of composite fibers to CSDX and shows that when DX is present, the pH decreases more quickly. This fact may be explained because DX has an acidic pH of 2–3 [49], and its chemical structure is composed of NH_2_ (which remains neutral) and OH (which has a negative charge), allowing DX to interact with HA and CS [29,50], both of which may cause a faster drop in pH.

The DX release values are shown in Figure 5, which show that the CSDX fibers have a burst release during the first 6 h and then no more DX is released. Significant differences in the total amount released as well as the release rate can be observed when HA is used. The release rate of CSHA10DX was similar, with a burst release during the first 6 h, but the DX amount released was more than double that of the case with no HA. CSHA50DX released an even higher amount of DX, and the release rate was different, showing a burst release during the first 6 h but followed by a more controlled release up to 72 h. In this case, the amount released triples the CSDX and doubles CSHA10DX. This could be explained by an interaction between positively charged CS and negatively charged HA and DX particles, implying a remarkable affinity between CS, HA, and DX with an intermolecular attraction [51]. As a result, more DX is encapsulated within HA-containing fibers, resulting in a higher total amount released. Moreover, the differences in the release rate of CSHA50DX compared to the other two samples may be due to DX retention by HA particles, primarily via Van der Waals forces [46,52], making it a more durable and controlled drug release.

The CSHA75DX release rate follows the same pattern as the CSHA50DX, with a burst release in the first 6 h and then a controlled release up to 72 h. However, the total amount released is nearly half of what it should be because of the DX retention by HA, as previously explained. This could be due to HA particle saturation, which prevents the release of the encapsulated DX from the fibers.

DX was used at a concentration of 5 mg/mL [46]. The minimum inhibitory concentration (MIC) of DX is known to be ≤16 µg/mL, but this varies depending on the bacteria strain [53]. In all groups studied in our experiments, we obtained a DX release with an MIC in accordance with these data. We assumed a high DX loss during NaOH immersion, which was the reason behind our choosing a higher DX concentration as our first study.

We investigated CSHA biomaterial as a potential carrier of DX. We are confident that introducing these fibers into maxillary bone defects will benefit both pathogen elimination and bone regeneration. The release can be fine-tuned by incorporating varying amounts of HA as well as different concentrations of DX, with the optimal option determined by each individualized case.

## 4. Conclusions

We created a drug–polymer–bioceramic conjugate, which is easy to manipulate and tune for use as a multifunctional, regulated drug delivery for maxillary bone regeneration treatments. Results have shown that several parameters need to be optimized in order to obtain the chitosan fibers. The incorporation of HA enabled enhanced manipulability as well as smaller sized fibers and a more neutral pH. Furthermore, the release of bioactive molecules, such as DX, showed better release in the presence of the HA. This was especially important for preventing bacterial infections in the oral cavity and making CSHADX fibers a promising material for tissue regeneration.

Further research needs to be performed in order to have a more homogenous HA particle size which eventually will allow a better injectability that will then allow the preparation of 3D printed substrates. Combined with this, the crosslinking solution should be further optimized to allow a milder condition. This must be obtained by combining the chitosan with other polymers. The designed carriers may eventually be further explored as a dual drug delivery system based on the two different materials that compose the fibers. The research will be further improved in the short term by quantifying the effect in vitro.

## Figures and Tables

**Figure 1 materials-16-00685-f001:**
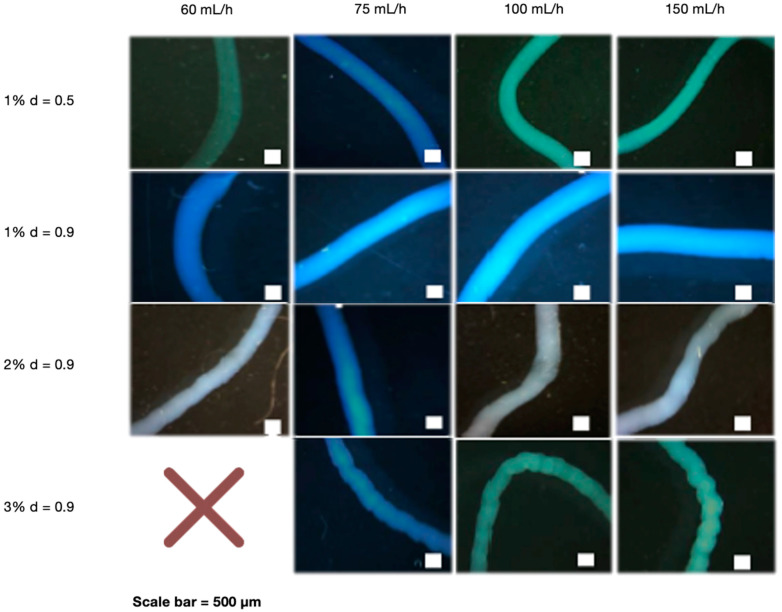
Optical microscope images of CS fibers at different percentages (1, 2 and 3%) and at different rates (60, 75, 100 and 150 mL/h). CS fibers at 1% were manufactured with 0.5- and 0.9-mm needle diameter, and the rest of fibers with 0.9-mm needle diameter. The NaOH solution was 0.5 M. Scale bar 500 µm.

**Figure 2 materials-16-00685-f002:**
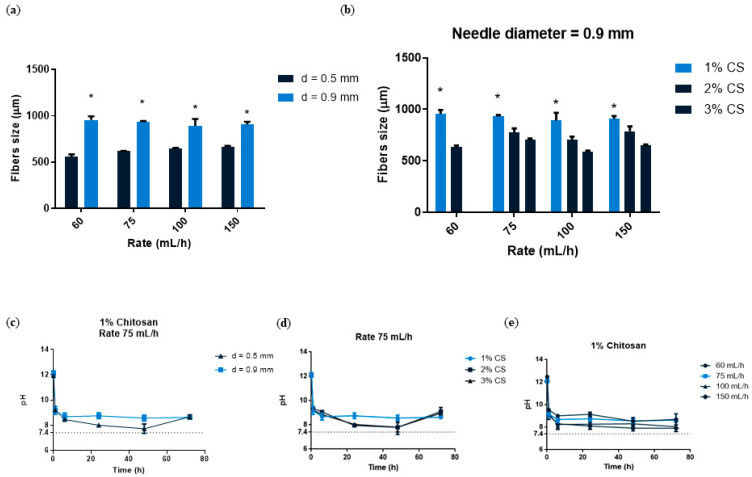
(**a**) CS fibers size at 1, 2 and 3% concentration fabricated with a needle diameter 0.9 mm and at different rates (60–150 mL/h). (**b**) CS fibers size with the two needle diameters (0.5 and 0.9 mm) and at different rates (60–150 mL/h). Data are shown as mean ± standard deviation (*n* = 6). (* *p* < 0.05). (**c**) pH analysis of 1% CS at a rate of 75 mL/h and with different needle diameters (0.5 or 0.9 mm). (**d**) pH analysis of different CS concentrations (1–3%) at a constant rate of 75 mL/h (*n* = 3). (**e**) pH analysis of 1% CS at different rates (60–150 mL/h) (*n* = 3).

**Figure 3 materials-16-00685-f003:**
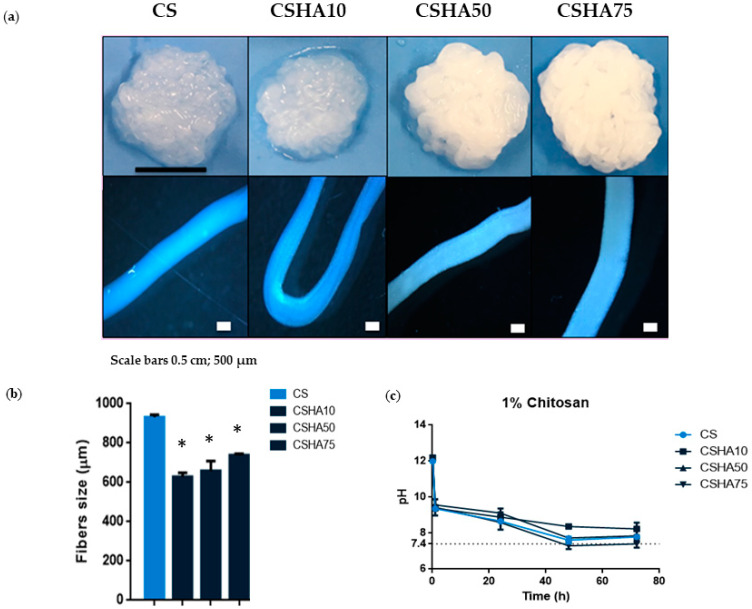
(**a**) Digital photographs and optical microscope images of 1% CS fibers manufactured with a needle diameter of 0.9 mm and at a constant rate of 75 mL/h mixed with different concentrations of HA (0–75%). Scale bars 0.5 cm and 500 µm. (**b**) CSHA fibers size with different HA concentrations (0–75%). Data are shown as mean ± standard deviation (*n* = 6). (* *p* < 0.05) (**c**) pH analysis of 1% CS at a constant rate of 75 mL/h with different HA concentrations (0–75%) (*n* = 3).

**Figure 4 materials-16-00685-f004:**
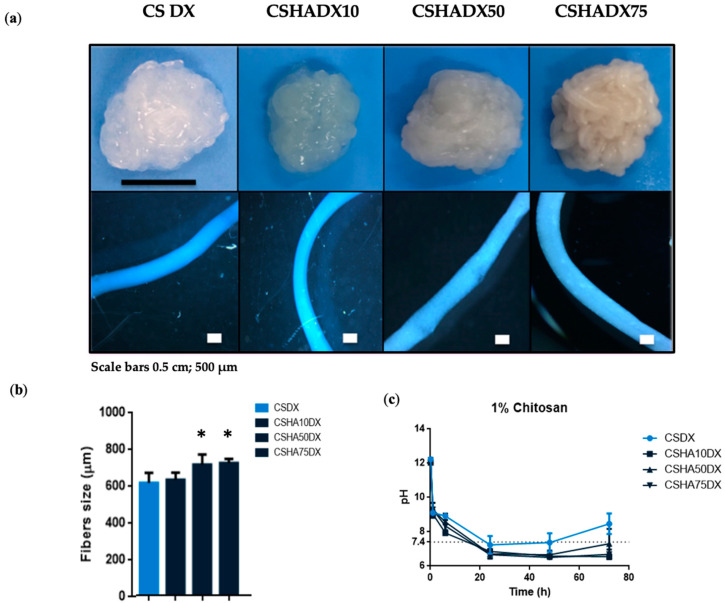
(**a**) Digital photographs and optical microscope images of 1% CS fibers manufactured with a needle diameter of 0.9 mm and at a constant rate of 75 mL/h mixed with different concentrations of HA (0–75%) with DX incorporation. Scale bars 0.5 cm and 500 µm. (**b**) CSHA fibers size with DX incorporation. Data are shown as mean ± standard deviation (*n* = 6). (* *p* < 0.05). (**c**) pH analysis of 1% CS at a constant rate of 75 mL/h with different HA concentrations (0–75%) and DX addition (*n* = 3).

**Figure 5 materials-16-00685-f005:**
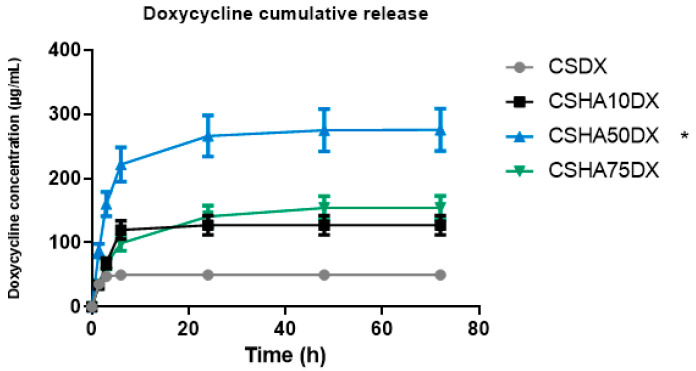
Cumulative amount of doxycycline (DX) released, and percentage of DX released as a function of time for samples 1% chitosan (CS) with different amounts of hydroxyapatite (HA) from 0 to 75%. (* *p* < 0.05).

## Data Availability

All data included in manuscript.

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
