# Peer review of "A Novel Chitosan Composite Biomaterial with Drug Eluting Capacity for Maxillary Bone Regeneration"

_materials, 2023, doi:10.3390/ma16020685_

Round 1
Reviewer 1 Report
Dear Auhtors,
thank you for an interesting article to review. Here are some suggestions. of mine how to improve it:
1. Introduction, sentence 1... are blood and bones transplants? I think I would use another word - please, transform the sentence
2. In the first paragraph, the authors write about tooth loss due to aging. I think it strongly depends on the life style, oral health and mental health - please try to add 1-2 sentences about it to expand this thought, see the reference:
Malicka B, Skośkiewicz-Malinowska K, Kaczmarek U. The impact of socioeconomic status, general health and oral health on Health-Related Quality of Life, Oral Health-Related Quality of Life and mental health among Polish older adults. BMC Geriatr. 2022 Jan 3;22(1):2. doi: 10.1186/s12877-021-02716-7. PMID: 34979959; PMCID: PMC8722217.
3. Please, note and add the information that natural polymers are used for oral health maintenance and that it is a main stream in todays medicine, see the ref.:
Paradowska-Stolarz A, Wieckiewicz M, Owczarek A, Wezgowiec J. Natural Polymers for the Maintenance of Oral Health: Review of Recent Advances and Perspectives. Int J Mol Sci. 2021 Sep 25;22(19):10337. doi: 10.3390/ijms221910337.
I would start with that general information, than add the detailed one about the use of polymers in bone regeneration procedures.
4. At the end of the introduction / and in the discussion, you should add the information on the side effects of tetracyclines, especially in refer to the dentition (please, find the references) and additionally, refer to other antibiotics used in medicine for bone and gingival regeneration, among them the most common one - metronidazole, see:
Kida, D.; Karolewicz, B.; Junka, A.; Sender-Janeczek, A.; Duś, I.; Marciniak, D.; Szulc, M. Metronidazole-Loaded Porous Matrices for Local Periodontitis Treatment: In Vitro Evaluation and In Vivo Pilot Study. Appl. Sci. 2019, 9, 4545. https://doi.org/10.3390/app9214545
5. The materials and metodology are well prepared
6. Although I am not a native speaker, not all of the sentences in your paper are clear. I think it is due to the "shortcuts" you are using. Please, go through the paper with native speaker, so that each sentence is clear for the reader - especially the one not knowing the subject of chitosan-use (eg. 1st paragraph, 1st sentence, as mentioned in details before). The paper should be constructed so that the unexperienced reader could gain as much as possible from it
7. Please, make separate chapters - results and discussion
8. The results section should have graphs and / or tables in it representing the most important results.
9. In the discussion, the authors should add the novel method of bone regeneration measurement. I think that it could be applied for the future studies by the Authors, see the ref:
- Jurczyszyn K, Kubasiewicz-Ross P, Nawrot-Hadzik I, Gedrange T, Dominiak M, Hadzik J. Fractal dimension analysis a supplementary mathematical method for bone defect regeneration measurement. Ann Anat. 2018 Sep;219:83-88. doi: 10.1016/j.aanat.2018.06.003.
10. The limitation section should be added
Although the paper is very interesting, it has some flaws. I would like the Authors to apply all the changes I mentioned above
Author Response
Dear Reviewers,
Thank you for all your valuable suggestions and comments about our article. We have modified the text according to the comments and these are shown in track changes.
Reviewer 1
- Introduction, sentence 1... are blood and bones transplants? I think I would use another word - please, transform the sentence.
Thank you for the comment. We have rewritten the sentence: “Bone grafting is one of the most performed operations.”
- In the first paragraph, the authors write about tooth loss due to aging. I think it strongly depends on the life style, oral health and mental health - please try to add 1-2 sentences about it to expand this thought, see the reference: Malicka B, Skośkiewicz-Malinowska K, Kaczmarek U. The impact of socioeconomic status, general health and oral health on Health-Related Quality of Life, Oral Health-Related Quality of Life and mental health among Polish older adults. BMC Geriatr. 2022 Jan 3;22(1):2. doi: 10.1186/s12877-021-02716-7. PMID: 34979959; PMCID: PMC8722217.
We have written in the text the main causes of tooth failure. Despite life style clearly affects the outcome of tooth behavior as well as aging, our introduction links in a much more natural way with tooth loss related with periodontal disease, caries and trauma. The following text is incorporated in the manuscript.
“The main reason is because of the increase of tooth loss in the population due to caries, periodontal disease, dental trauma or tumors [4], which cause alveolar bone loss compromising dental implant success”.
- Please, note and add the information that natural polymers are used for oral health maintenance and that it is a main stream in todays medicine, see the ref.:
Paradowska-Stolarz A, Wieckiewicz M, Owczarek A, Wezgowiec J. Natural Polymers for the Maintenance of Oral Health: Review of Recent Advances and Perspectives. Int J Mol Sci. 2021 Sep 25;22(19):10337. doi: 10.3390/ijms221910337.
I would start with that general information, than add the detailed one about the use of polymers in bone regeneration procedures.
We have added that general information and incorporated the reference as well.
“Autografts are the gold standard for bone regeneration, but they have limited availability. Recently, due to the search for alternative sources for bone regeneration, there has been an increase in interest in the use of natural polymers [5–7]. Collagen, hyaluronic acid, alginate and chitosan are some of the most studied natural polymers for bone regeneration. Moreover, they are being used in medicine and more specifically for the oral health maintenance. One of the main reasons is its biocompatibility, which they can induce a specific response of the body without affecting the immune system [8].”
Ref. 8. Paradowska-Stolarz A, Wieckiewicz M, Owczarek A, Wezgowiec J. Natural Polymers for the Maintenance of Oral Health: Review of Recent Advances and Perspectives. Int J Mol Sci. 2021 Sep 25;22(19):10337. doi: 10.3390/ijms221910337.
- At the end of the introduction / and in the discussion, you should add the information on the side effects of tetracyclines, especially in refer to the dentition (please, find the references) and additionally, refer to other antibiotics used in medicine for bone and gingival regeneration, among them the most common one - metronidazole, see:
Kida, D.; Karolewicz, B.; Junka, A.; Sender-Janeczek, A.; Duś, I.; Marciniak, D.; Szulc, M. Metronidazole-Loaded Porous Matrices for Local Periodontitis Treatment: In Vitro Evaluation and In Vivo Pilot Study. Appl. Sci. 2019, 9, 4545. https://doi.org/10.3390/app9214545.
Regarding tetracyclines side effects: The literature says: “One of the side-effects of this group of substances is their incorporation into tissues that are calcifying at the time of their administration.”
“These agents tend to remain in ossification zones for some time after systemic administration”.
This bone grafts would be placed in adults that have complete their calcification process both in teeth and in bone tissue. Moreover, this bone grafts would be administrated locally, and the concentration of the antibiotic released is expected to be minimal compared with the systemic administration.
Referring to other antibiotics used for bone and gingival regeneration, such as metronidazole, and regarding the paper that you have cited to us, as we have mentioned in our article, we have chosen doxycycline not only because its antibacterial effects but also for its bone regenerative effects. “Tetracyclines have been claimed to be the only type of antibiotic that stimulate bone mineralization”.
Sánchez AR, Rogers RS 3rd, Sheridan PJ. Tetracycline and other tetracycline-derivative staining of the teeth and oral cavity. Int J Dermatol. 2004 Oct;43(10):709-15. doi: 10.1111/j.1365-4632.2004.02108.x. PMID: 15485524.
- The materials and metodology are well prepared.
We appreciate your comment.
- Although I am not a native speaker, not all of the sentences in your paper are clear. I think it is due to the "shortcuts" you are using. Please, go through the paper with native speaker, so that each sentence is clear for the reader - especially the one not knowing the subject of chitosan-use (eg. 1st paragraph, 1st sentence, as mentioned in details before). The paper should be constructed so that the unexperienced reader could gain as much as possible from it.
The paper has been revised by a native speaker.
- Please, make separate chapters - results and discussion.
Thank you for your suggestion. Initially we considered having two separate sections. Nevertheless, following the below described guide for authors for the journal, both sections can be combined and hence we decided to combine both in order to make the manuscript with a more natural flow. As mentioned in the section of “Manuscript preparation” in the journal Materials, Discussion section may be combined with Results:
“Discussion: Authors should discuss the results and how they can be interpreted in perspective of previous studies and of the working hypotheses. The findings and their implications should be discussed in the broadest context possible and limitations of the work highlighted. Future research directions may also be mentioned. This section may be combined with Results.”
- The results section should have graphs and / or tables in it representing the most important results.
Graphs and tables are incorporated in the text.
- In the discussion, the authors should add the novel method of bone regeneration measurement. I think that it could be applied for the future studies by the Authors, see the ref:
Jurczyszyn K, Kubasiewicz-Ross P, Nawrot-Hadzik I, Gedrange T, Dominiak M, Hadzik J. Fractal dimension analysis a supplementary mathematical method for bone defect regeneration measurement. Ann Anat. 2018 Sep;219:83-88. doi: 10.1016/j.aanat.2018.06.003.
We think it is a very interesting novel method for bone regeneration. Nevertheless, we think it could be mentioned and used in a future article with in vivo experiments.
- The limitation section should be added.
The limitations has been added at the Results and Discussion section:
“Further research needs to be performed in order to have a more homogenous HA particle size which eventually will allow a better injectability that will eventually allow preparing 3D printed substrates. Combined with this, the crosslinking solution should be further optimized to allow a milder condition. This must be obtained by eventually combining the chitosan with other polymers. The designed carriers may eventually be further explored as a dual drug delivery system based on the two different materials that compose the fibers. The research will be further improved in short term by quantifying the effect in vitro.
Reviewer 2 Report
The article entitled “A novel chitosan composite biomaterial with drug eluting capacity for maxillary bone regeneration”. The authors elaborated an approach to produce chitosan-based fiber. The article is interesting and the experiments are accurately described. I suggest this article can be accepted after some modifications.
1. Please provide the manufacture of chitosan.
2. How to collect the chitosan fiber in NaOH solution?
3. There is no reference for measuring of amount of DX.
4. Section 3.1: the description about chitosan (the first paragraph) should be moved to Introduction.
5. Does this chitosan fiber contain sodium acetate?
6. What is kind of interactions to induce the formation of chitosan fiber?
Author Response
Dear Reviewers,
Thank you for all your valuable suggestions and comments about our article. We have modified the text according to the comments and these are shown in track changes.
- Please provide the manufacture of chitosan.
The chitosan manufacturing has been incorporated in the text as follows:
“CS fibers were designed as follows: Fibers were prepared in order to either form fiber-based scaffolds for tissue engineering or to determine the optimum conditions for future use in 3D printing. First, CS (Sigma-Aldrich) was prepared by dissolving 1 mL of glacial acetic acid (CH3COOH) (Panreac) into 99 mL of deionized water (DI water, from the Milli-Q water system) and stirring at 800 rpm. The solution was then treated with varying amounts of chitosan, ranging from 1 to 3 g, and stirred at 450 rpm for one hour at room temperature, to produce 1 to 3% wt chitosan solutions.”
- How to collect the chitosan fiber in NaOH solution?
This information has been added at the end of the first paragraph of Materials and methods section: “The samples were fabricated in triplicates. The collection of fibers from the NaOH solution was made with tweezers.”
- There is no reference for measuring of amount of DX.
For measuring the amount of DX, the DX absorption spectra was measured, and the result of the maximum wavelength was 351nm.
“The amount of DX released was measured using a spectrophotometer by measuring the absorbance at 351 nm (Bio-Tek Synergy HT) [35].”
This is the reference that corroborates it: Khampieng T, Wnek GE, Supaphol P. Electrospun DOXY-h loaded-poly(acrylic acid) nanofiber mats: in vitro drug release and antibacterial properties investigation. J Biomater Sci Polym Ed. 2014;25(12):1292-305. doi: 10.1080/09205063.2014.929431. Epub 2014 Jun 19. PMID: 24945329.
- Section 3.1: the description about chitosan (the first paragraph) should be moved to Introduction.
The description about Chitosan has been moved to the Introduction section:
“There are several options for treating maxillary bone loss. Enhancing maxillary bone regeneration with biomaterials is one of them, and has gained popularity in recent years, particularly the use of bioceramics, which are thought to be a good option due to the properties mentioned above. However, because every maxillary bone defect is different in shape and size, using a bioceramic materials alone may be difficult for the clinician to adapt correctly. Because of its easy tailorable properties, a natural polymer has been chosen as the matrix of the composite. Furthermore, they have excellent biocompatibility. Among natural polymers, chitosan has grown in popularity in tissue engineering (FDA approval for wound dressings [29]) and drug delivery. We chose it to synthesize our fibers because it is a biodegradable, non-toxic, and mucoadhesive biomaterial [30], with a low-cost and sustainable processing.”
- Does this chitosan fiber contain sodium acetate?
These chitosan fibers do not contain sodium acetate.
- What is kind of interactions to induce the formation of chitosan fiber?
The interactions to induce the formation of chitosan fiber are as follows. This information has been incorporated in the introduction.
“Crosslinking is a physiochemical technique in which a charged polymer, in this case CS, is forming intermolecular chemical bonds between chains of the polymer, gaining stability and resistance [7]. CS crosslinking to form the fiber shape was obtained with a solution of sodium hydroxide (NaOH).”
Reviewer 3 Report
I was pleased to review the article ID materials-2119647 entitled “A novel chitosan composite biomaterial with drug eluting capacity for maxillary bone regeneration” for the Materials Journal. The article investigates a novel biomaterial to be used in bone regeneration. Overall the article is poorly organized and presents several missing and confusing information that significantly decreases the power for publication in such a high-impact journal:
The main flaw is that there is no single in vivo or in vitro (Cells) evaluation to state that the material is suitable for maxillary bone regeneration.
Abstract:
Provide further details about “In vitro characterization of the biomaterial was performed.”
All groups must be mentioned and properly defined.
The absence of a clear conclusion was noted.
Introduction:
The last paragraph of the introduction must be rewritten. One paragraph alone describing the DX, and then the purpose of the study.
Methodology:
To allow reproducibility, the author cannot say “different amounts of chitosan, ranging from 1 to 3g”; they must state precise concentrations.
The same applies to “Different parameters were varied”.
Again for, “different smounts of HA were incroproated within the chitosan mixture.”
Missing reference: “DX was selected to be incorporated into the CSHA composite because it is a broad spectrum antibiotic and is known to have an osteogenic regeneration capacity.”
Provide a reference for the wavelength used: “The amount of DX released was measured using a spectrophotometer by measuring the absorbance at 351 nm”.
Conclusion: The sentence is extensive. The vast majority should be placed at the beginning of the discussion. Then, provide a short and clear conclusion in the appropriate section.
“Drug delivery for maxillary bone regeneration treatments.”: The analyses performed in the present study allow the authors to affirm that? Has any maxillary-specific methodology been employed?
Missing Author Contributions, Funding information, Institutional Review Board Statement, Data Availability Statement, and Acknowledgments.
Author Response
Dear Reviewers,
Thank you for all your valuable suggestions and comments about our article. We have modified the text according to the comments and these are shown in track changes.
- Abstract:
Provide further details about “In vitro characterization of the biomaterial was performed.”
All groups must be mentioned and properly defined.
Further details have been added and all groups have been mentioned and defined:
“The use of fibers made of 1, 2 and 3% of chitosan (CS) and 10, 50 and 75% of hydroxyapatite (HA) loaded with an antibiotic (doxycycline, DX) and fabricated with the help of an injection pump is presented as a new strategy for improving maxillary bone regeneration. In vitro characterization of the DX controlled released from the fibers was quantified after mixing the different amounts of HA.”
The absence of a clear conclusion was noted.
It was rewritten and added last sentences to provide a conclusion
“When compared to pristine chitosan fibers, the hydroxyapatite concentration dictated the combined fast and controlled release profile of DX, which simultaneously reduced the pH below physiological conditions. This was especially important in preventing bacterial infections in the oral cavity, which is one of the main common reasons for bone grafting failure. Hence, CSHADX fibers are a promising candidate graft material for enhancing bone tissue regeneration in dental clinical practice”
- Introduction:
The last paragraph of the introduction must be rewritten. One paragraph alone describing the DX, and then the purpose of the study.
The last paragraph has been rewritten:
“Doxycycline (DX) is a tetracycline antibiotic with a broad spectrum of action that has been used to treat bacterial infections of the oral cavity [32]. Besides that, tetracyclines have been reported to be the only antibiotics that stimulate bone mineralization [33,34].
This antibiotic was therefore combined with the CS-HA composite biomaterial in order to stimulate maxillary bone regeneration. For this reason, we propose a simple method involving the use of an injection pump to fabricate chitosan fibers with the incorporation of HA and DX. Hence, the goal of this study is to characterize the various fibers obtained and to study its DX controlled release while comparing different percentages of HA.”
- Methodology:
To allow reproducibility, the author cannot say “different amounts of chitosan, ranging from 1 to 3g”; they must state precise concentrations.
The precise concentrations have been described:
“The solution was then treated with varying amounts of chitosan, at amounts of 1, 2 or 3 g, and stirred at 450 rpm for one hour at room temperature, to produce 1, 2 and 3% wt chitosan solutions.”
The same applies to “Different parameters were varied”.
The different parameters studied have been described:
“Different parameters such as the molarity of the crosslinking solution, the CS concentration, the HA concentration, the needle diameter and the injection speed, were varied to obtain fibers with different sizes, textures, and consistencies. The syringe was then placed in an injection pump (kd Scientific pump KDS-200-CE) which was then used to extrude the fibers at different constant injection speeds of 60, 75, 100 or 150 mL/h using a needle diameter of 0.5 or 0.9 mm. The fibers were extruded into a NaOH solution of 0.05 M, 0.1 or 0.5 M which was used as the crosslinking agent.”
Again for, “different amounts of HA were incorporated within the chitosan mixture.”
The different amounts of HA have been described:
“The composite fibers were prepared in a similar way to those described with the pristine chitosan fibers. In this case, different amounts of HA (25%, 50% and 75%) were incorporated into the chitosan mixture.”
- Missing reference: “DX was selected to be incorporated into the CSHA composite because it is a broad spectrum antibiotic and is known to have an osteogenic regeneration capacity.”
The references have been added:
“DX was selected to be incorporated into the CSHA composite because it is a broad-spectrum antibiotic and is known to have an osteogenic regeneration capacity [32,33].”
- Provide a reference for the wavelength used: “The amount of DX released was measured using a spectrophotometer by measuring theabsorbance at 351 nm”.
The reference have been added:
For measuring the amount of DX, the DX absorption spectra was measured, and the result of the maximum wavelength was 351nm.
“The amount of DX released was measured using a spectrophotometer by measuring the absorbance at 351 nm (Bio-Tek Synergy HT) [35].”
This is the reference that corroborates it: Khampieng T, Wnek GE, Supaphol P. Electrospun DOXY-h loaded-poly(acrylic acid) nanofiber mats: in vitro drug release and antibacterial properties investigation. J Biomater Sci Polym Ed. 2014;25(12):1292-305. doi: 10.1080/09205063.2014.929431. Epub 2014 Jun 19. PMID: 24945329.
- Conclusion: The sentence is extensive. The vast majority should be placed at the beginning of the discussion. Then, provide a short and clear conclusion in the appropriate section.
We have rewritten this section
We created a drug-polymer-bioceramic conjugate, easy to manipulate and tune for use as multifunctional regulated drug delivery for maxillary bone regeneration treatments. Results have shown that several parameters need to be optimized in order to obtain the chitosan fibers. The incorporation of HA enabled enhanced manipulability as well smaller sized fibers as well as a more neutral pH. Furthermore, the release of bioactive molecules, such as DX, showed better release in the presence of the HA. This was especially important for preventing bacterial infections in the oral cavity and making CSHADX fibers a promising material for tissue regeneration.
- “Drug delivery for maxillary bone regeneration treatments.”: The analyses performed in the present study allow the authors to affirm that? Has any maxillary-specific methodology been employed?
We have rewritten the sentence:
“We have designed a drug-polymer-bioceramic conjugate, easy to manipulate and tune for use as multifunctional regulated drug delivery for a potential use in maxillary bone regeneration treatments.”
- Missing Author Contributions, Funding information, Institutional Review Board Statement, Data Availability Statement, and Acknowledgments.
The information is incorporated in the manuscript.
Round 2
Reviewer 1 Report
Dear Authors,
Thank you for the corrections. Please, correspond to the other types of antibiotics, metronidazole. Please add the separate chapter of limitations (not added to the discussion).
Author Response
Dear reviewer,
We thank you once again for taking the time to review our manuscript. We have followed your indications and have incorporated the changes with track changes.
Comments:
Thank you for the corrections. Please, correspond to the other types of antibiotics, metronidazole. Please add the separate chapter of limitations (not added to the discussion).
We thank you for your positive feedback. We have incorporated the sentence regarding the metronidazole:
Besides that, tetracyclines, together with other antibiotics such as metronidazole, have been reported to be the only antibiotics that stimulate bone mineralization [33,34].
Furthermore, we have moved the limitations of the study out of the discussion and is now together with the conclusions.
Reviewer 3 Report
The authors substantially improved the manuscript.
Author Response
Thank you for your positive feedback.